# Redundancy Aware Multiple Reference Based Gainwise Evaluation of Extractive Summarization

## Abstract

While very popular for evaluating extractive summarization task, the ROUGE metric has long been criticized for its lack of semantic awareness and its ignorance about the ranking quality of the summarizer. Thanks to previous research that has addressed these issues by proposing a gain-based automated metric called *Sem-nCG*, which is both rank and semantic aware. However, *Sem-nCG* does not consider the amount of redundancy present in a model-generated summary and currently does not support evaluation with multiple reference summaries. Unfortunately, addressing both these limitations simultaneously is not trivial. Therefore, in this paper, we propose a redundancy-aware *Sem-nCG* metric and demonstrate how this new metric can be used to evaluate model summaries against multiple references. We also explore different ways of incorporating redundancy into the original metric through extensive experiments. Experimental results demonstrate that the new redundancy-aware metric exhibits a higher correlation with human judgments than the original *Sem-nCG* metric for both single and multiple reference scenarios.

## 1 Introduction

For the past two decades, ROUGE Lin (2004b) has been the most used metric for evaluating extractive summarization tasks. Nonetheless, ROUGE has long been criticized for its lack of semantic awareness Graham (2015); Ng & Abrecht (2015); Ganesan (2018); Yang et al. (2018) and its ignorance about the ranking quality of the extractive summarizer Akter et al. (2022).

To address these issues, previous work has proposed a gain-based metric called *Sem-nCG* Akter et al. (2022) to evaluate extractive summaries by incorporating rank and semantic awareness. Redundancy, a crucial factor in evaluating extractive summaries, was not, however, included in the *Sem-nCG* metric. Additionally, their proposed *Sem-nCG* metric does not support the evaluation of model summaries against multiple references. However, it is well recognized that a set of documents can have multiple, very different, and equally valid summaries; as such, obtaining multiple reference summaries can improve the stability of the evaluation Nenkova (2005); Lin (2004a). Unfortunately, addressing both these limitations simultaneously is not trivial, and a systematic study of how to incorporate redundancy and multiple references into the existing *Sem-nCG* metric is duly warranted.

In this paper, we first incorporate redundancy into the previously proposed *Sem-nCG* metric. In other words, we propose a redundancy-aware *Sem-nCG* metric by exploring different ways of incorporating redundancy into the original metric. Through extensive experiments, we demonstrate that the redundancy-aware *Sem-nCG* exhibits a notably stronger correlation with humans than the original *Sem-nCG* metric.

Next, we demonstrate how this redundancy-aware metric could be applied to evaluate model summaries against multiple references. This is a non-trivial task because *Sem-nCG* evaluates a model-generated summary by considering it as a ranked list of sentences and then comparing it against an automatically inferred *ground-truth* ranked list of sentences within a source document based on a single human written summary Akter et al. (2022). However, in the case of multiple references, the *ground-truth* ranked list of source sentences must be inferred based on all available human-written reference summaries, not just one.

When multiple reference summaries are available, the traditional way of computing ROUGE/BERTScore is to compute the corresponding metric score for each reference and then average those scores. While this is certainly possible for *Sem-nCG* too, it is problematic for the following two reasons: 1) Multiple ground-truth rankings will need to be created, one for each reference summary available, which is computationally very expensive, and 2) Human-written summary qualities differ not only in writing style but also in focus and including multiple reference summaries with a lot of terminology variations and paraphrase make the automated evaluation metric less stable Cohan & Goharian (2016). Therefore, we opted to infer a single/unique ground-truth ranking based on multiple reference summaries in this work.

The problem of inferring a unique ground-truth ranking based on multiple reference summaries can be framed in many ways; e.g., one way to solve this problem is to infer ranks based on each reference and then aggregate them; another option is to merge multiple references into a single reference (a non-trivial task) and then infer the ranks of the source sentences. In this work, we have explored multiple ways of inferring ground-truth ranks to facilitate the evaluation using multiple references. Our findings suggest that, compared to the conventional ROUGE and BERTScore metric, the redundancy-aware *Sem-nCG* exhibits a stronger correlation with human judgments for evaluating model summaries when multiple references are available. Therefore, we encourage the community to use redundancy-aware *Sem-nCG* to evaluate extractive summarization tasks.

## 2 Related Work

The most common method for evaluating model summaries has been to compare them against human-written reference summaries. ROUGE Lin (2004b) considers direct lexical overlap and afterwards different version of ROUGE Graham (2015) has also been proposed including *ROUGE* with word embedding Ng & Abrecht (2015) and synonym Ganesan (2018), graph-based lexical measurement ShafieiBavani et al. (2018), Vanilla *ROUGE* Yang et al. (2018) and highlight-based *ROUGE* Hardy et al. (2019) to mitigate the limitations of original ROUGE. Metrics based on semantic similarity between reference and model summaries have also been proposed to capture the semantics, including S+WMS Clark et al. (2019), MoverScore Zhao et al. (2019), and BERTScore Zhang et al. (2020). Reference-free evaluation has also been a recent trend to avoid dependency on human reference Böhm et al. (2019); Peyrard (2019); Sun & Nenkova (2019); Gao et al. (2020); Wu et al. (2020).

Despite the fact that the *extractive* summarizing task is typically framed as a sentence ranking problem, none of the metrics mentioned above evaluate the quality of the ranker. To address this issue, Recently Akter et al. (2022) has proposed a rank-aware and gain-based evaluation metric for extractive summarization called *Sem-nCG*, but it does not incorporate redundancy and also lacks evaluation with multiple references, which are two significant limitations that need to be addressed and hence, the focus of this work.

Redundancy in extracted sentences is a prominent issue in extractive summarization systems. Maximal Marginal Relevance (MMR) Carbonell & Goldstein (1998) is a classic algorithm to penalize redundancy in model summary. There are several approaches that explicitly model redundancy and use algorithms to avoid selecting sentences that are too similar to those that have already been extracted Ren et al. (2016). Trigram blocking Paulus et al. (2018) is another popular approach to reduce redundancy in model summary. Chen et al. (2021) has shown how to compute self-referenced redundancy score while evaluating the model summary. In this work, we explore various ways to incorporate redundancy into the original *Sem-nCG* metric.

In the context of multi-reference summary evaluation, our work is additionally distinctive since we do not follow the conventional procedure of computing the evaluation metric for each reference separately and then estimating their average/max. Instead, we use a variety of human-written reference summaries to infer a single, unified ground-truth ranked list of source sentences, after which the sem-nCG score is computed only once.

When multiple reference summaries are available, Researchers have also suggested Pyramid-based Nenkova & Passonneau (2004) approaches for summary evaluation. However, since the pyramid must be manually constructed and requires more manual labor, this method received little attention. Although the method has undergone numerous improvements Passonneau et al. (2013); Yang et al. (2016); Shapira et al. (2019);

Mao et al. (2020), it still needs a substantial amount of manual effort, making it unsuitable for large-scale evaluation.

Recently, for NLG evaluation unified framework Deng et al. (2021); Zhong et al. (2022) to predict different aspects of the generated text has been proposed. Even though these metrics can be applied to text summarization, it is still a data-driven approach where the pseudo-data generation approach is erroneous and it is unclear why the model produces such scores.

## 3  Methodology

***Sem-nCG*** **Score:** Normalized Cumulative Gain ($nCG$) is a popular evaluation metric in information retrieval to evaluate the quality of a ranker. nCG compares the model ranking with an *ideal* ranking and assigns a certain score to the model based on some pre-defined gain. Akter et al. (2022) has utilized the idea of $nCG$ in the evaluation of extractive summarization. The basic concept of Sem-nCG is to compute the gain ($CG@k$) obtained by a top $k$ extracted sentences and divide that by the maximum/ideal possible gain ($ICG@k$), where the gains are inferred by comparing the input document against a human written summary. Mathematically:

$$Sem\text{-}nCG@k = \frac{CG@k}{ICG@k} \tag{1}$$

**Redundancy Score:** We followed Chen et al. (2021) to compute self-referenced redundancy score. The summary, $X$, itself is used as the reference to determine the degree of semantic similarity between each summary token/sentence and the other tokens/sentences. The average of maximum semantic similarity is used to determine the redundancy score. For a given summary, $X = \{x_1, x_2, ..., x_n\}$, the calculation is as follows:

$$Score_{\text{red}} = \frac{\sum_i max_{j:i\neq j} Sim(x_j, x_i)}{|\mathbf{X}|} \tag{2}$$

where, $j : i \neq j$ denotes that the similarity between $x_i$ and itself has not been considered. Note that $Score_{red} \in [0, 1]$ in our case and lower is better.

**Final Score:** We used the following formula to calculate the final score after obtaining the scores of *Sem-nCG* and $Score_{red}$:

$$Score = \lambda * Sem\text{-}nCG + (1 - \lambda) * (1 - Score_{red}) \tag{3}$$

Here, $\lambda \in [0, 1]$ is a hyper-parameter to scale the weight between $Score_{red}$ and *Sem-nCG*. *Score* $\in [0, 1]$ where higher score means better summary.

## 4  Experimental Setup

### 4.1  Dataset

Human correlation is an essential attribute to consider while assessing the quality of a metric. To compute the human correlation of the new redundancy-aware *Sem-nCG* metric, we utilized SummEval dataset from Fabbri et al. (2021)[1]. The annotations include summaries generated by 16 models (abstractive and extractive) from 100 news articles (1600 examples in total) on the CNN/DailyMail Dataset. Each source news article includes the original CNN/DailyMail reference summary as well as 10 additional crowd-sourced reference summaries. Each summary was annotated by 5 independent crowd-sourced workers and 3 independent experts (8 annotations in total) along the four dimensions: *Consistency*, *Relevance*, *Coherence* and *Fluency* Fabbri et al. (2021). As this work focuses on the evaluation of extractive summarization, we considered the output generated by extractive models and filtered out samples comprising less than 3 sentences (as we report *Sem-nCG@3*). Additionally, we considered the expert annotations for the meta-evaluation, as non-expert annotations can be risky Gillick & Liu (2010).

---

[1]`https://github.com/Yale-LILY/SummEval`

### 4.1.1 Human Evaluation Components

To calculate the Kendall's Tau ($\tau$) rank correlation for the redundancy-aware *Sem-nCG* metric, we used four quality dimensions following Akter et al. (2022); Fabbri et al. (2021).

**Consistency:** refers to the fact that the contents in the summary and the source are the same. Only assertions from the source are included in factually consistent summaries, which do not include any trippy facts.

**Relevance:** getting the most important information from a source. The annotators were to penalize summaries with redundancy and excessive information. In the summary, only important information from the source should be included.

**Coherence:** overall summary sentence quality while keeping a coherent body of information on a topic rather than a tangle of related information Dang (2005).

**Fluency:** the structure and quality of the summary sentences. As mentioned in Dang (2005) "should have no formatting problems, capitalization errors or obviously ungrammatical sentences (e.g., fragments, missing components) that make the text difficult to read."

As was done in Akter et al. (2022), for each sample, from the 11 available reference summaries, we considered 3 settings: Less Overlapping Reference/LOR (highly abstractive references with fewer lexical overlap with the original document), Medium Overlapping Reference/MOR (medium lexical overlap with the original document) and Highly Overlapping Reference/HOR (highly extractive references with high lexical overlap with the original document).

## 4.2 Embedding for Groundtruth Ranking

The core of the *Sem-nCG* metric is to automatically create the groundtruth/ideal ranking against which the model ranking is compared. To create the groundtruth ranking, Akter et al. (2022) used various sentence embeddings. Similarly, we utilized various sentence embeddings as well since our goal is to compare the new redundancy-aware *Sem-nCG* metric to the original *Sem-nCG* metric. Specifically, we considered Infersent (v2) Conneau et al. (2017), Semantic Textual Similarity benchmark (STSb - bert/roberta/distilbert) Reimers & Gurevych (2019), Elmo Peters et al. (2018) and Google Universal Sentence Encoder (USE) Cer et al. (2018) with enc-2 Iyyer et al. (2015) based on the deep average network, to infer the groundtruth/ideal ranking of the sentences within the input document with guidance from the human written summaries.

### 4.2.1 Details of Sentence Embedding Used

**Infersent Conneau et al. (2017):** Infersent-v2 is trained with fastText word embedding and generates 4096-dimensional sentence embedding using a BiLSTM network with max-pooling.

**Elmo Peters et al. (2018):** The contextualized word embedding was transformed into a sentence embedding using a fixed mean-pooling of all contextualized word representations with embedding shape 1024.

**Google Universal Sentence Encoder (USE) Cer et al. (2018):** We utilized USE with enc-2 Iyyer et al. (2015) which is based on the deep average network to transform input text to a 512-dimensional sentence embedding.

**Semantic Textual Similarity benchmark (STSb) Reimers & Gurevych (2019):** Sentence Transformer allows to generate dense vector representations of sentences. Three of the best available models that were optimized for semantic textual similarity were considered: STSb-bert (embedding size 1024), STSb-roberta (embedding size 1024) and STSb-distilbert (embedding size 768).

## 4.3 *Score_red* Computation

To compute the self-referenced redundancy score, we used the top-3 sentences from the model generated summary (as we report *Sem-nCG@3*). We calculated each sentence's maximum similarity to other sentences and then averaged it to get the desired $Score_{red}$. We experimented with four distinct variations to compare

the sentences: cosine similarity (by converting sentences to STSb-distilbert Reimers & Gurevych (2019) embeddings), ROUGE Lin (2004b), MoverScore Zhao et al. (2019) and BERTScore Zhang et al. (2020).

### 4.3.1 Explanation of Metrics for *Score$_{red}$*

**ROUGE Lin (2004b):** Between the generated summary and reference summary, ROUGE counts the overlap of textual units (n-grams, word sequences).

**MoverScore Zhao et al. (2019):** uses the Word Mover's Distance Kusner et al. (2015) to calculate the semantic distance between a summary and a reference text, pooling n-gram embedding from BERT representations.

**BERTScore Zhang et al. (2020):** calculates similarity scores by matching generated and reference summaries on a token level. The cosine similarity between contextualized token embeddings from BERT is maximized by computing token matching greedily.

**Cosine Similarity:** Sentences are converted to sentence embedding using STSb-distilbert Reimers & Gurevych (2019). Then the semantic similarity of sentences is measured using cosine similarity between sentence vectors.

The code for the metrics used can be found here.[2]

## 5 Results

### 5.1 Redundancy-aware *Sem-nCG*

We first considered how redundancy-aware Sem-nCG performs in extractive summarization with single reference. As shown in Table 1, we computed Kendall's tau ($\tau$) correlation between the expert given score for model summary and the Sem-nCG score with/without redundancy along the four meta-evaluation criteria: *Consistency*, *Relevance*, *Coherence*, and *Fluency*, for different embedding variations (to create the groudtruth ranking) and different approaches to compute *Score$_{red}$*. We utilized Equation 3 to compute the redundancy-aware *Sem-nCG* score, where lambda ($\lambda$) is a hyper-parameter choice and is set to $\lambda = 0.5$ empirically. In Table 1 w/o redundancy refers to Equation 1.

Table 1 shows that the redundancy-aware *Sem-nCG* metric outperforms the original *Sem-nCG* metric in terms of *Consistency*, *Relevance*, and *Coherence*; with a 5% improvement in *Relevance* and a 14% improvement in *Coherence* for less overlapping references (LOR). We also observe improvements in the *Relevance* (9%) and *Coherence* (20%) dimensions for medium overlapping references (MOR). For High Overlapping References (HOR), the improvement is 8% and 22% for *Relevance* and *Coherence*, respectively.

We also observe that STSb-distilbert embedding is a better choice in the *Consistency* dimension, whereas USE with enc-2 is a better choice in the *Relevance* and *Coherence* dimensions to construct the groundtruth ranking. Therefore, we recommend STSb-distilbert to create groundtruth ranking if *Consistency* is a top priority, otherwise, we recommend using USE with enc-2. A groundtruth ranking was also created by combining STSb-distilbert and USE into an ensemble, which showed balanced performance across all four dimensions. It also appears that ROUGE and BERTScore provide comparable performances while computing *Score$_{red}$*. However, using ROUGE score as self-referenced redundancy will be a better choice as evident from Section 5.3.

In Table 2 Kendall's tau correlation of ROUGE and BERTScore has been demonstrated to get an idea of the advantage of redundancy-aware *Sem-nCG* and it is clearly evident that redundancy-aware Sem-nCG also exhibits stronger correlation than these metrics. Table 3 shows a qualitative example for the evaluation of a model-extracted summary.

---

[2]https://github.com/Yale-LILY/SummEval/tree/master/evaluation/summ_eval

| Embedding | Type | Consistency | | | Relevance | | | Coherence | | | Fluency | | |
|---|---|---|---|---|---|---|---|---|---|---|---|---|---|
| | | LOR | MOR | HOR | LOR | MOR | HOR | LOR | MOR | HOR | LOR | MOR | HOR |
| Inferesent | w/o redundancy | 0.08 | 0.06 | 0.08 | 0.07 | 0.12 | 0.09 | 0.06 | 0.06 | 0.04 | **0.05** | 0.03 | **0.12** |
| + Redundancy penalty | Cosine Similarity | 0.04 | 0.02 | 0.06 | 0.08 | 0.15 | 0.13 | 0.14 | 0.19 | 0.18 | 0.02 | -0.02 | 0.08 |
| | ROUGE | 0.07 | 0.05 | 0.11 | 0.11 | 0.18 | 0.17 | 0.18 | 0.25 | **0.26** | -0.01 | -0.04 | 0.05 |
| | MoverScore | 0.05 | 0.06 | 0.11 | 0.09 | 0.15 | 0.12 | 0.11 | 0.13 | 0.11 | 0.03 | 0.01 | 0.11 |
| | BERTScore | 0.05 | 0.02 | 0.08 | 0.13 | 0.19 | 0.18 | 0.18 | 0.22 | 0.24 | -0.01 | -0.04 | 0.04 |
| Elmo | w/o redundancy | 0.06 | 0.07 | 0.09 | 0.02 | 0.08 | 0.06 | 0.02 | 0.02 | 0.01 | 0.00 | 0.01 | 0.06 |
| + Redundancy penalty | Cosine Similarity | 0.03 | 0.03 | 0.05 | 0.04 | 0.13 | 0.10 | 0.12 | 0.14 | 0.14 | -0.06 | -0.05 | 0.02 |
| | ROUGE | 0.08 | 0.05 | 0.08 | 0.07 | 0.15 | 0.14 | 0.17 | 0.20 | 0.20 | -0.06 | -0.06 | 0.01 |
| | MoverScore | 0.08 | 0.07 | 0.10 | 0.04 | 0.10 | 0.09 | 0.07 | 0.06 | 0.06 | -0.02 | -0.01 | 0.05 |
| | BERTScore | 0.06 | 0.03 | 0.05 | 0.09 | 0.17 | 0.16 | 0.17 | 0.19 | 0.18 | -0.06 | -0.07 | 0.00 |
| STSb-bert | w/o redundancy | 0.11 | 0.08 | 0.09 | 0.03 | 0.13 | 0.12 | -0.01 | 0.06 | 0.01 | 0.03 | **0.10** | 0.03 |
| + Redundancy penalty | Cosine Similarity | 0.08 | 0.01 | 0.06 | 0.05 | 0.17 | 0.13 | 0.10 | 0.18 | 0.16 | -0.05 | 0.02 | 0.05 |
| | ROUGE | 0.12 | 0.05 | 0.09 | 0.08 | **0.22** | 0.18 | 0.14 | 0.25 | 0.22 | -0.04 | -0.04 | 0.01 |
| | MoverScore | 0.12 | 0.06 | 0.10 | 0.05 | 0.16 | 0.15 | 0.04 | 0.11 | 0.09 | -0.01 | 0.02 | 0.08 |
| | BERTScore | 0.10 | 0.01 | 0.06 | 0.11 | **0.22** | **0.20** | 0.14 | 0.24 | 0.20 | -0.06 | -0.04 | 0.01 |
| STSb-roberta | w/o redundancy | 0.12 | **0.14** | 0.07 | 0.07 | 0.07 | 0.05 | 0.04 | 0.00 | -0.02 | -0.01 | 0.01 | 0.06 |
| + Redundancy penalty | Cosine Similarity | 0.09 | 0.07 | 0.05 | 0.08 | 0.11 | 0.06 | 0.13 | 0.13 | 0.10 | -0.06 | -0.05 | -0.01 |
| | ROUGE | 0.12 | 0.11 | 0.09 | 0.11 | 0.16 | 0.10 | 0.18 | 0.20 | 0.17 | -0.07 | -0.07 | -0.04 |
| | MoverScore | 0.13 | 0.13 | 0.10 | 0.09 | 0.10 | 0.07 | 0.08 | 0.07 | 0.04 | -0.03 | 0.00 | 0.04 |
| | BERTScore | 0.10 | 0.08 | 0.05 | 0.13 | 0.18 | 0.12 | 0.17 | 0.18 | 0.15 | -0.08 | -0.06 | -0.04 |
| USE | w/o redundancy | 0.05 | 0.04 | 0.04 | 0.11 | 0.14 | 0.08 | 0.07 | 0.08 | 0.02 | 0.03 | 0.05 | 0.08 |
| + Redundancy penalty | Cosine Similarity | 0.02 | -0.01 | 0.03 | 0.10 | 0.16 | 0.09 | 0.16 | 0.19 | 0.16 | -0.05 | 0.01 | 0.03 |
| | ROUGE | 0.06 | 0.02 | 0.07 | 0.13 | 0.21 | 0.14 | 0.20 | **0.26** | 0.23 | -0.06 | 0.00 | 0.00 |
| | MoverScore | 0.07 | 0.03 | 0.07 | 0.13 | 0.16 | 0.11 | 0.13 | 0.13 | 0.10 | 0.01 | 0.03 | 0.06 |
| | BERTScore | 0.03 | -0.01 | 0.05 | **0.15** | 0.22 | 0.17 | **0.21** | 0.24 | 0.22 | -0.06 | 0.00 | 0.00 |
| STSb-distilbert | w/o redundancy | 0.17 | 0.09 | **0.12** | 0.06 | 0.09 | 0.07 | 0.06 | 0.03 | -0.01 | 0.01 | 0.03 | 0.04 |
| + Redundancy penalty | Cosine Similarity | 0.16 | 0.04 | 0.06 | 0.07 | 0.12 | 0.07 | 0.14 | 0.16 | 0.11 | -0.05 | -0.03 | -0.04 |
| | ROUGE | 0.16 | 0.06 | 0.08 | 0.10 | 0.16 | 0.12 | 0.17 | 0.21 | 0.17 | -0.06 | -0.04 | -0.05 |
| | MoverScore | **0.18** | 0.08 | 0.10 | 0.08 | 0.12 | 0.09 | 0.09 | 0.09 | 0.04 | -0.02 | 0.01 | 0.01 |
| | BERTScore | 0.14 | 0.03 | 0.05 | 0.12 | 0.18 | 0.14 | 0.17 | 0.20 | 0.16 | -0.06 | -0.05 | -0.05 |
| Ensemble$_{sim}$ | w/o redundancy | 0.12 | 0.08 | 0.07 | 0.10 | 0.12 | 0.07 | 0.08 | 0.06 | 0.00 | 0.01 | 0.04 | 0.05 |
| + Redundancy penalty | Cosine Similarity | 0.11 | 0.02 | 0.04 | 0.10 | 0.16 | 0.09 | 0.16 | 0.20 | 0.15 | -0.06 | -0.01 | -0.01 |
| | ROUGE | 0.13 | 0.05 | 0.08 | 0.13 | 0.21 | 0.14 | 0.20 | **0.26** | 0.21 | -0.05 | -0.03 | -0.03 |
| | MoverScore | 0.14 | 0.06 | 0.08 | 0.12 | 0.15 | 0.10 | 0.14 | 0.13 | 0.08 | -0.01 | 0.03 | 0.03 |
| | BERTScore | 0.10 | 0.03 | 0.05 | **0.15** | **0.22** | 0.16 | **0.21** | 0.25 | 0.20 | -0.05 | -0.02 | -0.03 |

Table 1: Kendall's tau ($\tau$) correlation coefficients of expert annotations for different embedding variations of *Sem-nCG* along with various redundancy penalties when $\lambda = 0.5$. Low overlapping reference (LOR), medium overlapping CNN/DailyMail reference (MOR), and high overlapping reference (HOR) were chosen from 11 reference summaries per example to demonstrate the correlation. The highest value in each column is in bold red.

| | Consistency | | | Relevance | | | Coherence | | | Fluency | | |
|---|---|---|---|---|---|---|---|---|---|---|---|---|
| | LOR | MOR | HOR | LOR | MOR | HOR | LOR | MOR | HOR | LOR | MOR | HOR |
| ROUGE-1 | 0.08 | 0.05 | 0.01 | 0.07 | 0.21 | 0.22 | 0.03 | 0.13 | 0.13 | 0.05 | 0.05 | 0.05 |
| ROUGE-L | 0.02 | 0.06 | -0.01 | 0.03 | 0.19 | 0.15 | -0.02 | 0.14 | 0.08 | 0.01 | 0.04 | -0.07 |
| BERTScore | 0.06 | 0.10 | 0.07 | 0.10 | 0.18 | 0.20 | 0.06 | 0.15 | 0.11 | 0.08 | 0.05 | 0.04 |

Table 2: Kendall's tau correlation coefficients of ROUGE and BERTScore for Low overlapping reference (LOR), medium overlapping CNN/DailyMail reference (MOR), and high overlapping reference (HOR) chosen from 11 reference summaries per example to demonstrate the correlation.

## 5.2 Hyperparameter Choice

In figure 1, we have varied $\lambda \in [0, 1]$ for the 3 scenarios (LOR, MOR and HOR) and computed human correlation along four dimensions (*Consistency*, *Relevance*, *Coherence* and *Fluency*) when different embeddings are used to create the groundtruth ranking and ROUGE score is used to compute $Score_{red}$. Human correlations with BERTScore-based redundancy are presented in figure 2. For both redundancy penalties, it shows that higher lambda ($\lambda \geq 0.6$) achieves better correlation for the *Consistency* dimensions, which makes sense because higher lambda means giving more weight to *Sem-nCG*. For *Relevance* and *Coherence* dimensions, a lower lambda ($\lambda$) value between $[0.3 - 0.5]$ is a better choice as lower $\lambda$ means more penalty

to redundancy. It appears that for *Fluency* all metric variations struggle. It is evident that $\lambda = 0.5$ gives comparable performance in all four quality dimensions (consistency, relevance, coherence and fluency) and thus we recommend using $\lambda = 0.5$ while adopting Equation 3 to compute redundancy-aware *Sem-nCG*.

| |
|---|
| **Article:** Last week she was barely showing – but Demelza Poldark is now the proud mother to the show's latest addition. Within ten minutes of tomorrow night's episode, fans will see Aidan Turner's dashing Ross Poldark gaze lovingly at his new baby daughter. As Sunday night's latest heartthrob, women across the country have voiced their longing to settle down with the brooding Cornish gentleman – but unfortunately, it seems as if his heart is well and truly off the market. Scroll down for the video. Last week she was barely showing – but Demelza Poldark is now the proud mother to the show's latest addition He may have married his red-headed kitchen maid out of duty, but as he tells her that she makes him a better man, audiences can have little doubt about his feelings. What is rather less convincing, however, is the timeline of the pregnancy. With the climax of the previous episode being the announcement of the pregnancy, it is quite a jump to the start of tomorrow's installment where Demelza, played by Eleanor Tomlinson, talks about being eight months pregnant. Just minutes after – once again without any nod to the passing of time – she is giving birth, with the last month of her pregnancy passing in less than the blink of an eye. With the climax of the previous episode being the announcement of the pregnancy, it is quite a jump to the start of tomorrow's instalment where Demelza, played by Eleanor Tomlinson, talks about being eight months pregnant As Sunday night's latest heartthrob, women across the country have voiced their longing to settle down with Poldark – but unfortunately, it seems as if his heart is well and truly off the market Their fast relationship didn't go unnoticed by fans. One posted on Twitter: 'If you are pregnant in Poldark times expect to have it in the next 10 minutes' It is reminiscent of the show's previous pregnancy that saw Elizabeth, another contender for Ross's affection, go to full term in the gap between two episodes. This didn't go unnoticed by fans, who posted on Twitter: 'Poldark is rather good, would watch the next one now. Though if you are pregnant in Poldark times expect to have it in the next 10 minutes. |
| **Model Summary:** Within ten minutes of tomorrow night's episode, fans will see aidan turner's dashing ross poldark gaze lovingly at his new baby daughter. Last week she was barely showing – but demelza poldark is now the proud mother to the show's latest addition. Last week she was barely showing – but demelza poldark is now the proud mother to the show's latest addition. (clearly redundant extractive summary) |
| **Score$_{\mathbf{red}}$ for model summary**: 0.40 |
| **Less Overlapping Reference (LOR)**: A celebrity recently welcomed a baby into the world and the wife discusses her experiences with her pregnancy. She has wanted to settle down for a while and is glad her pregnancy wasn't noticeable on television. |
| **Medium Overlapping/CNN Reference (MOR)**: SPOILER ALERT: Maid gives birth to baby on Sunday's episode. Only announced she was pregnant with Poldark's baby last week. |
| **High Overlapping Reference (HOR)**: In the latest episode, Demelza Poldark talks about being 8 months pregnant. Ross Poldark, who is off the market and in love with Demelza, will be shown gazing lovingly at his new baby daughter tomorrow night. |
| **Sem-nCG Score** only according to equation 1 for
LOR: 0.67      MOR: 0.733      HOR: 0.8 |
| **Revised Sem-nCG Score** along with Score$_{\mathrm{red}}$ according to equation 3 for $\lambda = 0.5$
LOR: 0.532      MOR: 0.565      HOR: 0.599 |
| **Human Evaluation** (annotated by experts and score ranged between 0-1)
Coherence: 0.47      Consistency: 1      Fluency: 1      Relevance: 0.67 |

Table 3: An example of the model summary evaluation using the redundancy-aware Sem-nCG metric.

| Metric | Multi-Ref LOR, MOR, HOR | | | | Multi-Ref LORs | | | | Multi-Ref MORs | | | | Multi-Ref HORs | | | |
|---|---|---|---|---|---|---|---|---|---|---|---|---|---|---|---|---|
| | Con | Rel | Coh | Flu | Con | Rel | Coh | Flu | Con | Rel | Coh | Flu | Con | Rel | Coh | Flu |
| ROUGE-1 | 0.00 | -0.01 | -0.09 | -0.01 | -0.05 | 0.05 | 0.00 | 0.01 | -0.05 | 0.09 | 0.04 | -0.01 | -0.02 | 0.21 | 0.13 | 0.10 |
| ROUGE-L | 0.00 | -0.01 | -0.09 | -0.01 | 0.00 | 0.04 | -0.01 | 0.01 | -0.06 | 0.07 | 0.04 | 0.00 | -0.01 | 0.15 | 0.09 | -0.04 |
| BERTScore | 0.09 | 0.19 | 0.14 | 0.03 | 0.01 | 0.07 | -0.01 | 0.04 | -0.04 | 0.05 | 0.03 | 0.05 | 0.04 | 0.20 | 0.12 | 0.06 |

Table 4: Kendall Tau ($\tau$) correlation coefficient for ROUGE and BERTScore for consistency (con), relevance (rel), coherence (coh) and fluency (flu) dimension for evaluating extractive model summaries with multiple references.

## 5.3 Redundancy-aware *Sem-nCG* for Evaluation with Multiple References

SummEval Fabbri et al. (2021) dataset contains 11 reference summaries. For summary evaluation with multiple references, we considered the lexical overlap of the reference summaries with the original document to demonstrate the terminology variations. Then we considered 3 less overlapping references as Multi-Ref LORs, 3 medium overlapping references as Multi-Ref MORs and 3 high overlapping references as Multi-Ref HORs. We have also mixed up 1 LOR, 1 MOR and 1 HOR and considered this set as Muti-Ref LOR, MOR, HOR to see how the evaluation metric correlates in different terminology variations.

| Embedding | Multi-Ref LOR, MOR, HOR | | | | | | | | | | | |
|---|---|---|---|---|---|---|---|---|---|---|---|---|
| | w/o Redundancy | | | | + Redundancy Penalty (ROUGE) | | | | + Redundancy Penalty (BERTScore) | | | |
| | Consistency | Relevance | Coherence | Fluency | Consistency | Relevance | Coherence | Fluency | Consistency | Relevance | Coherence | Fluency |
| Infersent | 0.07 | 0.11 | 0.08 | **0.06** | 0.11 | 0.18 | **0.27** | 0.01 | 0.09 | 0.18 | 0.20 | 0.03 |
| Elmo | 0.09 | 0.06 | 0.01 | 0.00 | 0.09 | 0.12 | 0.18 | -0.05 | 0.09 | 0.12 | 0.11 | -0.03 |
| STSb-bert | 0.10 | 0.12 | 0.04 | 0.06 | 0.09 | 0.19 | 0.24 | -0.02 | 0.10 | **0.20** | 0.18 | 0.01 |
| STSb-roberta | 0.14 | 0.10 | 0.01 | 0.02 | 0.12 | 0.17 | 0.21 | -0.06 | **0.13** | 0.17 | 0.13 | -0.02 |
| USE | 0.04 | 0.12 | 0.08 | 0.05 | 0.06 | 0.19 | 0.26 | -0.03 | 0.05 | 0.19 | 0.20 | 0.01 |
| STSb-distilbert | 0.14 | 0.13 | 0.05 | 0.02 | 0.10 | 0.19 | 0.23 | -0.04 | 0.12 | **0.20** | 0.17 | -0.01 |
| Embedding | Multi-Ref LORs | | | | | | | | | | | |
| | w/o Redundancy | | | | + Redundancy Penalty (ROUGE) | | | | + Redundancy Penalty (BERTScore) | | | |
| | Consistency | Relevance | Coherence | Fluency | Consistency | Relevance | Coherence | Fluency | Consistency | Relevance | Coherence | Fluency |
| Infersent | 0.03 | 0.10 | 0.09 | **0.07** | 0.05 | **0.16** | **0.25** | 0.02 | 0.02 | 0.15 | 0.18 | 0.04 |
| Elmo | 0.04 | 0.05 | -0.04 | 0.03 | 0.05 | 0.12 | 0.15 | -0.04 | 0.03 | 0.11 | 0.06 | -0.01 |
| STSb-bert | 0.08 | 0.10 | 0.02 | 0.01 | 0.09 | 0.15 | 0.20 | -0.06 | 0.06 | 0.15 | 0.13 | -0.04 |
| STSb-roberta | 0.10 | 0.07 | -0.04 | 0.00 | **0.11** | 0.15 | 0.17 | -0.07 | 0.09 | 0.15 | 0.09 | -0.04 |
| USE | 0.02 | 0.05 | 0.01 | 0.03 | 0.04 | 0.12 | 0.19 | -0.04 | 0.02 | 0.10 | 0.12 | 0.00 |
| STSb-distilbert | 0.10 | 0.04 | -0.02 | -0.02 | **0.11** | 0.09 | 0.15 | -0.09 | 0.09 | 0.09 | 0.09 | -0.07 |
| Embedding | Multi-Ref MORs | | | | | | | | | | | |
| | w/o Redundancy | | | | + Redundancy Penalty (ROUGE) | | | | + Redundancy Penalty (BERTScore) | | | |
| | Consistency | Relevance | Coherence | Fluency | Consistency | Relevance | Coherence | Fluency | Consistency | Relevance | Coherence | Fluency |
| Infersent | 0.08 | 0.08 | 0.03 | **0.06** | 0.10 | **0.15** | **0.23** | -0.02 | 0.08 | **0.15** | 0.16 | 0.02 |
| Elmo | 0.06 | 0.05 | -0.02 | 0.00 | 0.04 | 0.13 | 0.16 | -0.07 | 0.05 | 0.11 | 0.08 | -0.05 |
| STSb-bert | 0.07 | 0.05 | 0.02 | 0.01 | 0.09 | 0.13 | 0.22 | -0.08 | 0.07 | 0.12 | 0.15 | -0.04 |
| STSb-roberta | 0.05 | 0.07 | -0.01 | 0.02 | 0.07 | 0.14 | 0.21 | -0.07 | 0.04 | 0.14 | 0.14 | -0.03 |
| USE | 0.02 | 0.08 | 0.05 | 0.01 | 0.04 | 0.15 | **0.25** | -0.06 | 0.02 | 0.14 | 0.17 | -0.03 |
| STSb-distilbert | **0.11** | 0.01 | 0.00 | -0.01 | 0.09 | 0.07 | 0.17 | -0.10 | 0.10 | 0.06 | 0.10 | -0.05 |
| Embedding | Multi-Ref HORs | | | | | | | | | | | |
| | w/o Redundancy | | | | + Redundancy Penalty (ROUGE) | | | | + Redundancy Penalty (BERTScore) | | | |
| | Consistency | Relevance | Coherence | Fluency | Consistency | Relevance | Coherence | Fluency | Consistency | Relevance | Coherence | Fluency |
| Infersent | 0.07 | 0.08 | 0.05 | 0.03 | **0.11** | 0.16 | 0.23 | -0.02 | 0.07 | 0.15 | 0.15 | 0.01 |
| Elmo | 0.04 | 0.09 | 0.02 | 0.06 | 0.06 | 0.16 | 0.19 | 0.00 | 0.04 | 0.14 | 0.11 | 0.03 |
| STSb-bert | 0.08 | 0.11 | 0.04 | 0.05 | 0.12 | 0.18 | **0.24** | -0.03 | 0.08 | 0.18 | 0.16 | 0.01 |
| STSb-roberta | 0.10 | 0.09 | 0.01 | 0.03 | 0.14 | 0.17 | 0.22 | -0.04 | 0.10 | 0.16 | 0.13 | 0.00 |
| USE | 0.04 | 0.14 | 0.07 | 0.05 | 0.07 | 0.20 | **0.24** | -0.03 | 0.04 | **0.21** | 0.18 | 0.01 |
| STSb-distilbert | 0.08 | 0.09 | 0.02 | 0.05 | **0.11** | 0.15 | 0.22 | -0.03 | 0.09 | 0.15 | 0.14 | 0.02 |

Table 5: Kendall Tau ($\tau$) correlation coefficient for **Ensemble$_{sim}$** when lambda ($\lambda$) = 0.5 for consistency, relevance, coherence and fluency dimension without redundancy and when ROUGE and BERTScore is used as redundancy penalty for different terminology variations of multiple references (highly abstractive (LORs), medium overlapping (MORs) and highly extractive (HORs) references). The best value in each dimension has been bold red.

Table 4 confirms that ROUGE shows very poor correlation in all the dimensions (consistency, relevance, coherence, and fluency) in all the scenarios and shows slightly better correlation in Multi-Ref HORs (which is somewhat expected as ROUGE considers direct lexical overlap). Interestingly, BERTScore also shows poor correlation in all the settings supporting that the traditional evaluation metric becomes less stable for multiple reference summaries with lots of terminology variations Cohan & Goharian (2016).

In the original *Sem-nCG* metric, a groundtruth ranking is prepared by considering the cosine similarity between each sentence of the document and reference summary but the evaluation with multiple-reference was left as future work. As a starting point, how to incorporate multiple-reference summaries in the original *Sem-nCG* metric, we designed how to create the groundtruth ranking by considering multiple references. Here, we took the naive approach, first computing cosine similarity of each sentence of the document with each reference among multiple references. Then average it, which we called Ensemble$_{sim}$.

For Ensemble$_{rel}$, for each groundtruth ranking prepared for each reference among multiple reference summaries, we took the average of relevance (as it was computed in previously proposed *Sem-nCG* metric Akter et al. (2022)) and based on that we merged the groundtruth rankings into one groundtruth ranking. Then we use this groundtruth ranking to compute *Sem-nCG* for model extracted summary. With the original Sem-nCG metric, we have also incorporated redundancy into the *Sem-nCG* metric utilizing equation 3. We have only considered ROUGE and BERTScore as redundancy penalty both in Table 5 and 6 when $\lambda = 0.5$ (as evident from Section 5.2 that this setting gives better performance). We have also considered different embedding variations to create the groundtruth ranking.

From Table 5, we can see that redundancy-aware *Sem-nCG* shows better correlations for all the scenarios (multi-ref LORs, multi-ref MORs, multi-ref HORs and mixture of LOR, MOR & HOR). Both ROUGE and BERTScore provide comparable results for self-referenced redundancy penalties, with ROUGE score-

| Embedding | Multi-Ref LOR, MOR, HOR | | | | | | | | | | | |
|---|---|---|---|---|---|---|---|---|---|---|---|---|
| | w/o Redundancy | | | | + Redundancy Penalty (ROUGE) | | | | + Redundancy Penalty (BERTScore) | | | |
| | Consistency | Relevance | Coherence | Fluency | Consistency | Relevance | Coherence | Fluency | Consistency | Relevance | Coherence | Fluency |
| Infersent | 0.09 | 0.10 | 0.04 | 0.08 | 0.11 | 0.17 | 0.24 | 0.01 | 0.09 | 0.18 | 0.18 | 0.04 |
| Elmo | 0.09 | 0.06 | 0.02 | 0.01 | 0.09 | 0.13 | 0.20 | -0.05 | 0.09 | 0.12 | 0.13 | -0.03 |
| STSb-bert | 0.12 | 0.15 | 0.04 | 0.05 | 0.12 | 0.22 | 0.24 | -0.03 | 0.12 | 0.24 | 0.18 | 0.01 |
| STSb-roberta | **0.14** | 0.08 | 0.01 | 0.05 | 0.13 | 0.15 | 0.21 | -0.05 | 0.13 | 0.15 | 0.12 | -0.02 |
| USE | 0.04 | 0.16 | 0.11 | **0.08** | 0.05 | 0.21 | **0.29** | 0.00 | 0.04 | 0.22 | 0.24 | 0.05 |
| STSb-distilbert | 0.14 | 0.10 | 0.03 | 0.02 | 0.10 | 0.16 | 0.22 | -0.04 | 0.11 | 0.18 | 0.16 | -0.01 |

| Embedding | Multi-Ref LORs | | | | | | | | | | | |
|---|---|---|---|---|---|---|---|---|---|---|---|---|
| | w/o Redundancy | | | | + Redundancy Penalty (ROUGE) | | | | + Redundancy Penalty (BERTScore) | | | |
| | Consistency | Relevance | Coherence | Fluency | Consistency | Relevance | Coherence | Fluency | Consistency | Relevance | Coherence | Fluency |
| Infersent | 0.03 | 0.09 | 0.07 | **0.08** | 0.05 | 0.15 | **0.23** | 0.04 | 0.02 | 0.14 | 0.16 | 0.05 |
| Elmo | 0.03 | 0.04 | -0.04 | 0.03 | 0.04 | 0.10 | 0.14 | -0.03 | 0.03 | 0.09 | 0.06 | -0.01 |
| STSb-bert | 0.09 | 0.10 | 0.00 | 0.01 | 0.10 | **0.16** | 0.19 | -0.06 | 0.09 | 0.17 | 0.13 | -0.03 |
| STSb-roberta | 0.10 | 0.05 | -0.06 | 0.00 | 0.11 | 0.13 | 0.15 | -0.08 | 0.09 | 0.12 | 0.07 | -0.04 |
| USE | 0.04 | 0.08 | 0.03 | 0.04 | 0.05 | 0.14 | 0.22 | -0.04 | 0.03 | 0.13 | 0.15 | 0.01 |
| STSb-distilbert | **0.13** | 0.06 | 0.01 | -0.01 | 0.12 | 0.11 | 0.17 | -0.09 | 0.12 | 0.12 | 0.12 | -0.06 |

| Embedding | Multi-Ref MORs | | | | | | | | | | | |
|---|---|---|---|---|---|---|---|---|---|---|---|---|
| | w/o Redundancy | | | | + Redundancy Penalty (ROUGE) | | | | + Redundancy Penalty (BERTScore) | | | |
| | Consistency | Relevance | Coherence | Fluency | Consistency | Relevance | Coherence | Fluency | Consistency | Relevance | Coherence | Fluency |
| Infersent | 0.06 | 0.10 | 0.05 | 0.06 | 0.07 | 0.19 | **0.26** | -0.01 | **0.06** | 0.18 | **0.19** | 0.02 |
| Elmo | 0.06 | 0.06 | 0.00 | 0.02 | 0.04 | 0.13 | 0.17 | -0.06 | 0.04 | 0.12 | 0.11 | -0.02 |
| STSb-bert | 0.08 | 0.01 | -0.02 | 0.01 | **0.09** | 0.09 | 0.18 | -0.08 | 0.08 | 0.08 | 0.11 | -0.04 |
| STSb-roberta | 0.05 | 0.07 | 0.00 | 0.02 | 0.06 | 0.14 | 0.20 | -0.07 | 0.05 | 0.14 | 0.13 | -0.02 |
| USE | 0.01 | 0.09 | 0.05 | 0.01 | 0.04 | 0.16 | 0.24 | -0.05 | 0.01 | 0.16 | 0.19 | -0.02 |
| STSb-distilbert | 0.08 | 0.02 | 0.00 | -0.01 | 0.07 | 0.09 | 0.18 | -0.09 | 0.07 | 0.08 | 0.12 | -0.06 |

| Embedding | Multi-Ref HORs | | | | | | | | | | | |
|---|---|---|---|---|---|---|---|---|---|---|---|---|
| | w/o Redundancy | | | | + Redundancy Penalty (ROUGE) | | | | + Redundancy Penalty (BERTScore) | | | |
| | Consistency | Relevance | Coherence | Fluency | Consistency | Relevance | Coherence | Fluency | Consistency | Relevance | Coherence | Fluency |
| Infersent | 0.09 | 0.11 | 0.06 | 0.05 | 0.13 | 0.18 | 0.25 | -0.01 | 0.09 | 0.18 | 0.18 | 0.02 |
| Elmo | 0.05 | 0.08 | 0.02 | 0.05 | 0.07 | 0.16 | 0.19 | -0.01 | 0.05 | 0.14 | 0.12 | 0.02 |
| STSb-bert | 0.07 | 0.11 | 0.04 | 0.05 | 0.11 | 0.18 | 0.25 | -0.02 | 0.06 | 0.19 | 0.17 | 0.02 |
| STSb-roberta | 0.10 | 0.08 | 0.01 | 0.04 | **0.13** | 0.16 | 0.21 | -0.04 | 0.11 | 0.15 | 0.13 | 0.00 |
| USE | 0.06 | 0.13 | 0.07 | **0.05** | 0.09 | **0.20** | **0.26** | -0.02 | 0.06 | 0.20 | 0.19 | 0.02 |
| STSb-distilbert | 0.09 | 0.09 | 0.03 | 0.03 | 0.12 | 0.15 | 0.22 | -0.05 | 0.10 | 0.15 | 0.15 | 0.00 |

Table 6: Kendall Tau ($\tau$) correlation coefficient for **Ensemble$_{rel}$** when lambda ($\lambda$) = 0.5 for consistency, relevance, coherence and fluency dimension without redundancy and when ROUGE and BERTScore is used as redundancy penalty for different terminology variations of multiple references (highly abstractive (LORs), medium overlapping (MORs) and highly extractive (HORs) references). The best value in each dimension has been bold red.

based redundancy providing a marginally superior result. Interestingly, redundancy-aware *Sem-nCG* shows robust performance in all the scenarios while showing 25% improvement in coherence and 10% improvement in relevance dimension. Same patterns are observed when Ensemble$_{rel}$ is also used for the evaluation of multiple reference (See Table 6).

From our empirical evaluation, we would recommend USE embedding to create Ensemble$_{sim}$ (merging sentence-wise similarities across different references) with ROUGE redundancy penalty to evaluate extractive summary with multiple references.

# 6 Conclusion

Previous work has proposed the *Sem-nCG* metric exclusively for evaluating extractive summarization task considering both rank awareness and semantics. However, the *Sem-nCG* metric ignores redundancy in a model summary and does not support evaluation with multiple reference summaries, which are two significant limitations. In this paper, we have suggested a redundancy-aware multi-reference based *Sem-nCG* metric by exploring different embeddings and similarity functions which is superior compared to the previously proposed *Sem-nCG* metric along *Consistency*, *Relevance* and *Coherence* dimensions. Additionally, for summary evaluation using multiple references, we created a unique ground-truth ranking by incorporating multiple references rather than trivial max/average score computation with multiple references. Our empirical evaluation shows that the traditional metric becomes unstable when multiple references are available and the new redundancy-aware *Sem-nCG* shows a notably higher correlation with human judgments than ROUGE and BERTScore metric both for single and multiple references. Thus we encourage the community to evaluate extractive summaries using the new redundancy-aware *Sem-nCG* metric.

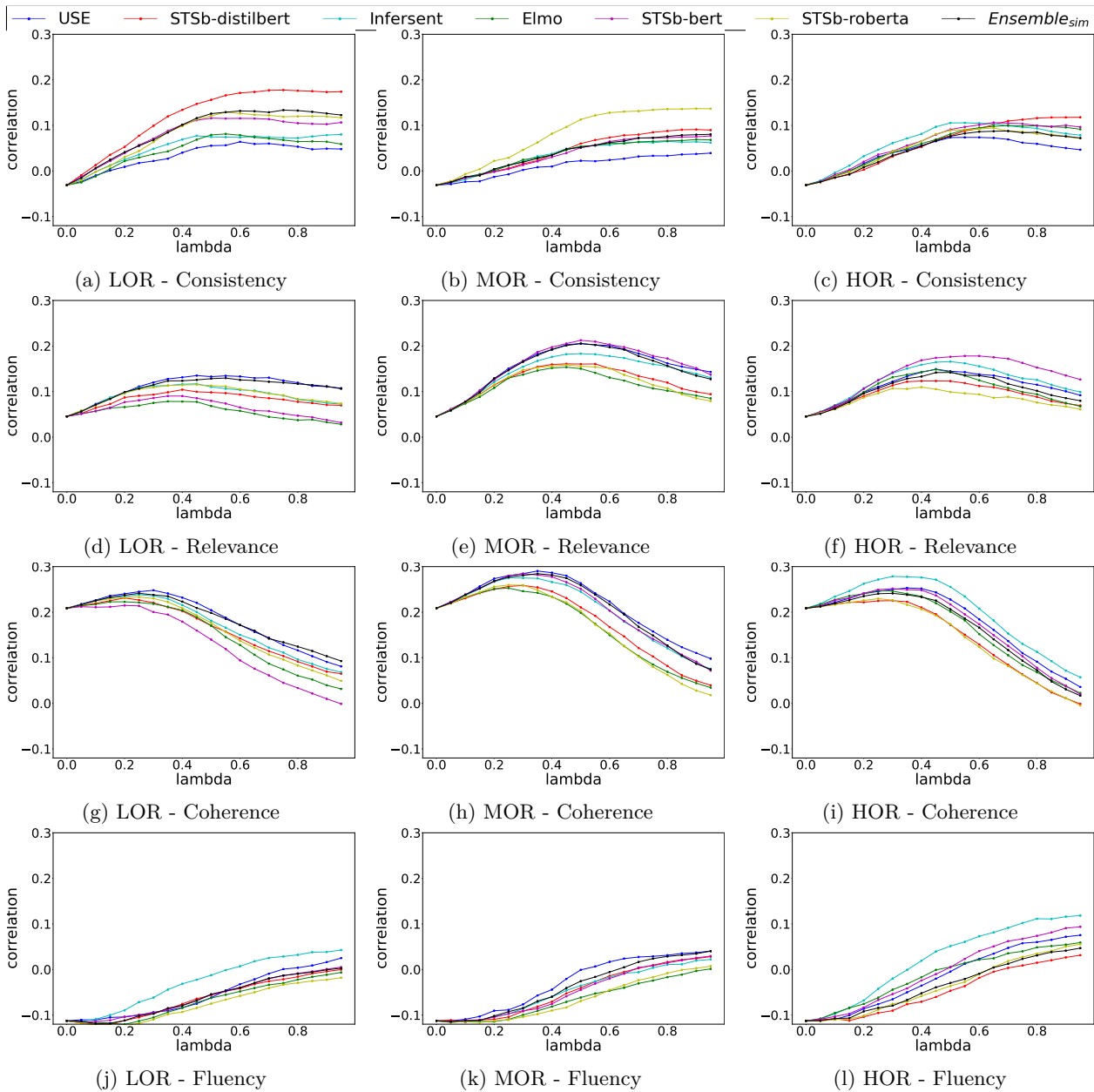

Figure 1: Kendall Tau ($\tau$) Correlation coefficient when lambda ($\lambda$) $\in [0,1]$ from (a)-(c) for Consistency, (d)-(f) for relevance, (g)-(i) for coherence and (j)-(l) for Fluency dimension when ROUGE score is used as redundancy penalty for less overlapping reference (LOR), medium overlapping reference (MOR) and high overlapping reference (HOR).

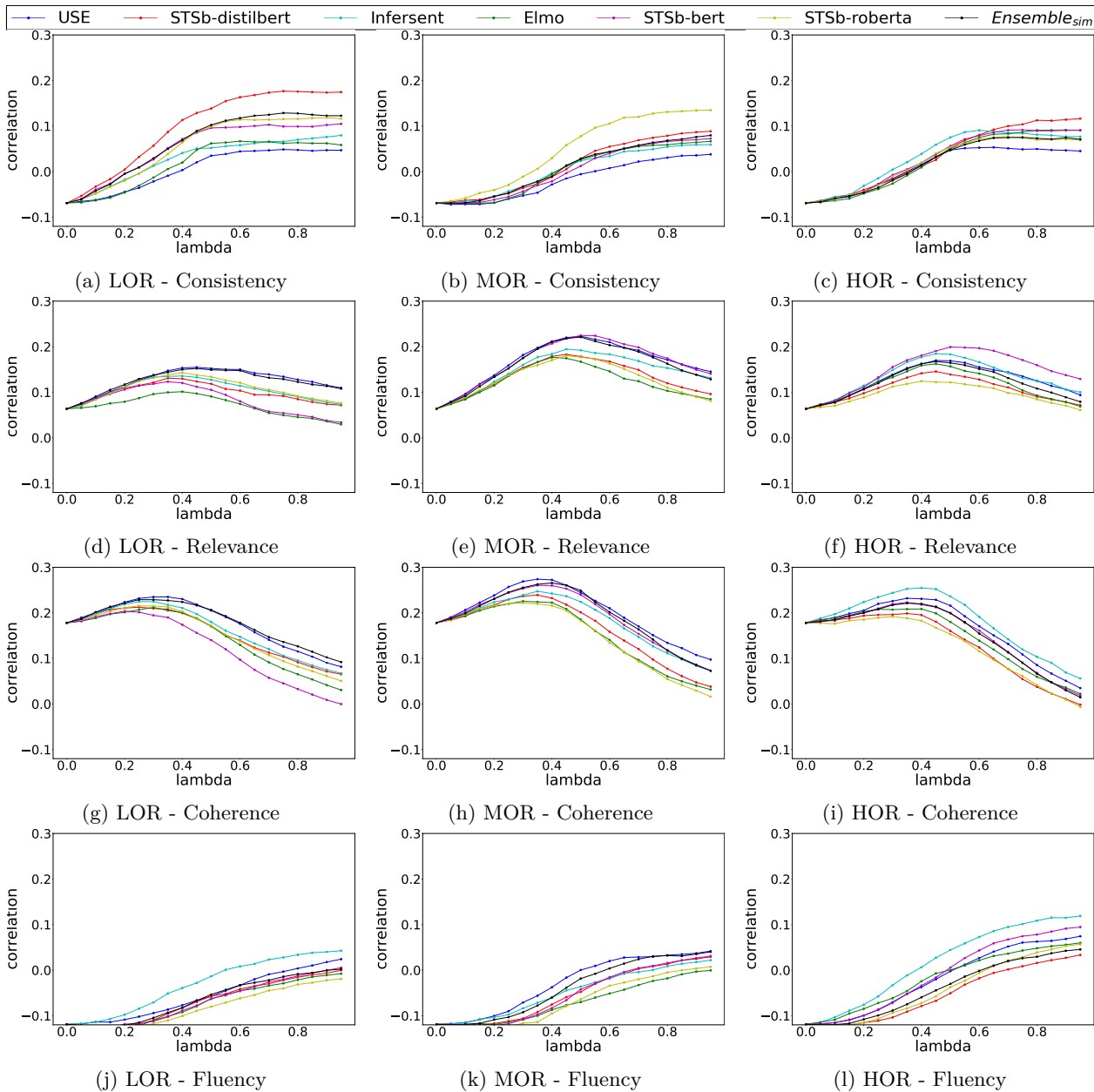

Figure 2: Kendall Tau ($\tau$) correlation coefficient when lambda ($\lambda$) $\in [0, 1]$ from (a)-(c) for consistency, (d)-(f) for relevance, (g)-(i) for coherence and (j)-(l) for fluency dimension when BERTScore is used as redundancy penalty for less overlapping reference (LOR), medium overlapping reference (MOR) and high overlapping reference (HOR).

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

# A  Appendix

# A  Appendix

## A.1  Computational Infrastructure & Runtime

| Computational Infrastructure | | |
|:---:|:---:|:---:|
| NVIDIA Quadro RTX 5000 GPUs | | |
| **Hyperparameter Search** | | |
| $\lambda \in [0,1]$ uniform-integer distribution | | |
| **Type** | **Variation** | **Runtime (s)** |
| | Cosine Similarity | 0.06 |
| $Score_{red}$ | ROUGE | 0.44 |
| | MoverScore | 0.23 |
| | BERTScore | 14.7 |
| | Infersent | 0.4 |
| | Elmo | 79.1 |
| | STSb-bert | 0.33 |
| $Sem\text{-}nCG$ | STSb-roberta | 0.34 |
| | USE | 20.2 |
| | STSb-distilbert | 0.13 |
| | $\text{Ensemble}_{\text{sim}}$ | 20.33 |

