# OpenReview forum: "Redundancy Aware Multiple Reference Based Gainwise Evaluation of Extractive Summarization"
_TMLR — Rejected by TMLR_

### Review · Reviewer_Y35R · 2023-09-08

**Summary Of Contributions:**

This paper proposes a new evaluation metric for extractive summarization. It extends Sem-nCG score to account for redundancy by adding a redundancy score from Chen et al 2021. The redundancy score measures the similarity between the sentences within a summary where more redundancy leads to a lower final score. The metric is evaluated along several axis from the SummEval dataset and shows to generally improve performance over Sem-nCG.

**Audience:**

Yes

**Claims And Evidence:**

No

**Requested Changes:**

Critical (from weaknesses above)
- Experiments on more recent models like pre-trained transformers showing the correlation with human judgment.
- Rewrite to make the metric more clear with all the components only having 1 choice

Strengthen
- More description of Sem-nCG score is needed. I had to go back to Atker et al (2022) to understand Sem-nCG. Given the importance of Sem-nCG in this paper, understanding Sem-nCG should be self-contained in this paper without having to reference other work. For example, how is gain measured? What is the maximum/ideal gain?
- From the intro, one of the main motivations for the new metric is handling multiple references and not being redundant. I don’t follow how this redundancy score handles multiple references. From my understanding, it seems to be computing the redundancy of a single summary, not multiple summaries.
- More emphasis needed in intro to clarify that this metric only applies to extractive summarization since this metric is ranking metric (which only works for summaries which are extracted from the document)

Questions
- I don't follow how there are 11 reference summaries for CNN/DM if it is an extractive summarization dataset? Does this mean there are 11 different sentences in the original document that can be consider a good summary?
- Table 1 says “The highest value in each column is in bold red.” - I only see highlights in green.
- Is the same embedding/similarity metric used to compute the ground-truth ranking and the redundancy score? In Table 1, what is the metric for computing the ground-truth ranking when there is no redundancy score?
- One of the motivations for this work is that “Multiple ground-truth rankings will need to be created, one for each reference summary available, which is computationally very expensive”. Why is getting a ground-truth ranking for each reference summary computationally very expensive? Isn’t it just a couple more cosine-similarity calls?

**Strengths And Weaknesses:**

Strength
- Adding redundancy to Sem-nCG seems to improve over Sem-nCG in certain settings

Weaknesses
- I think the experiments are lacking. The main benchmark used are the annotations from Fabbri et al 2021. The extractive models in Fabbi were from 2018 which were not Transformers and were not pre-trained. Given the use of transformers and pre-training for summarization, I don’t see the applicability of these results for today.
- The paper is proposing a metric, but it is not clear what the metric is since several components of the metric have various choices - 1) Which similarity function and 2) how to combine multiple references. This also makes the paper writing a bit hard to follow since the emphasis is on the various choices for the components instead of the metric.

---

> ### Author Response · Authors · 2023-09-14
>
> We genuinely appreciate your valuable time, prompt review, and the insightful comments you've provided. In the subsequent discussion, we will address various points and outline the revisions that will be implemented in response to your valuable suggestions.
>
> - “I think the experiments are lacking... results for today.”
>
> We used the dataset by Fabbri et al. (2021), the only available benchmark "meta-evaluation dataset" for extractive summarization, to the best of our knowledge. There is no human annotation dataset available for a more recent model's summary and we can’t do anything here. Sem-nCG's authors have demonstrated its correlation with human judgment on this dataset. To ensure a fair comparison, we maintained the same settings as the original Sem-nCG when assessing the redundancy-aware Sem-nCG.  To evaluate the transformer-based extractive summarization model, we will need a similar human-annotated dataset for the latest models with respect to the automated metric which is not in the scope of this paper. Alternatively, we can explore integrating recent embeddings into Sem-nCG and also use cosine similarity as a redundancy penalty to see if it increases the correlation with human judgment. Please let us know if you want to see these experiments in the revised version and we will be happy to report them.
>
> - “Experiments on more recent models like pre-trained transformers showing the correlation with human judgment.”
>
> Please see the above point. Even though “Liu et al. Revisiting the Gold Standard: Grounding Summarization Evaluation with Robust Human Evaluation. ACL (1) 2023” has published a new dataset, that work focuses mainly on how to increase human annotation reliability for summary evaluation with respect to Atomic Content Unit (ACU) and doesn’t provide human judgment for model’s summary along four summary quality dimensions: coherence, consistency, fluency and relevance.
>
> - I don't follow how there are 11 reference summaries for CNN/DM if it is an extractive summarization dataset? Does this mean there are 11 different sentences in the original document that can be consider a good summary?
>
> (Fabbri et al., 2021) has created a benchmark human annotation dataset called SummEval to evaluate the performance of summarization metrics with human judgment. Each source news article includes the original CNN/DailyMail reference summary as well as 10 additional crowd-sourced reference summaries. Any reference summaries among them can be used as a proxy to evaluate the system summary.  Each system/model summary was then annotated by 5 independent crowd-sourced workers and 3 independent experts (8 annotations in total) along the four dimensions: Consistency, Relevance, Coherence, and Fluency given these 11 reference summaries, with the hope that the automated metric to evaluate that system/model summary will have better alignment with human judgment along these four dimensions.
>
> - More description of the Sem-nCG score is needed...
>
> - More emphasis needed in intro ...
>
> We appreciate the reviewer’s suggestion and will include more clarity in the updated version
>
> - handling multiple references and not being redundant...
>
> To enhance Sem-nCG, we tackle its limitations by introducing two improvements: first, incorporating a redundancy penalty for system summaries, and second, how to create groundtruth rankings when multiple references are present. We consider redundancy in system summaries (as redundant system summary should be penalized) and use multiple references for the original Sem-nCG metric's groundtruth ranking.
>
> - Rewrite to make the metric more clear with all the components only having 1 choice
>
> Thank you so much for the suggestion. We also highlighted that in Sections 5.2 and 5.3. We thought that was clear but after reading your comment, we plan to include a summary table for the clarity of the hyperparameter choice.
>
> - Table 1 says “The highest value ... bold red.” - I only see highlights in green.
>
> Thank you so much for catching this typo. We will correct this in the updated version.
>
> - Is the same embedding/similarity metric used to compute the ground-truth ranking and the redundancy score? In Table 1, what is the metric for computing the ground-truth ranking when there is no redundancy score?
>
> We have experimented with different sentence embeddings to create the groundtruth. For example, Infersent, Elmo, Google Universal Sentence Encoder (USE) and STSb-bert, STSb-roberta, and STSb-distilbert to create the groundtruth ranking. On the other hand, as a redundancy penalty, we have used ROUGE score, Moverscore, BERTScore and cosine similarity (using stsb-distilbert embedding) (details in Section 4.2 and Section 4.3).
>
> Thank you once again for your time. We hope that the discussions outlined above adequately address any concerns you had, and all of these modifications will be incorporated into the revised version. Should you have any new questions or comments, please don't hesitate to comment.

---

### Review · Reviewer_iEg2 · 2023-09-26

**Summary Of Contributions:**

This paper proposes to integrate Sem-nCG, an evaluation metric for extractive summarization, with a redundancy score (Equation 3) to consider the amount of redundant information present in a model-generated summary. This paper also explores the use of multiple reference summaries. The authors claim that the proposed evaluation metric has a better correlation with human judgments on SummEval dataset for both single and multiple reference settings.

**Audience:**

Yes

**Broader Impact Concerns:**

None.

**Claims And Evidence:**

No

**Requested Changes:**

# Major comments

+ Reconsider the experiments and the presentations of the results.
+ Reconsider the descriptions so that findings and claims are backed up by the empirical results.
+ Reconsider the contribution of this work. For example, if this paper includes the method for handling multiple reference summaries as a *proposal*, we would expect to see the proposal in greater detail in Section 3 and empirical comparison with other methods (for considering multiple references) in Section 5.3.

# Minor comments

+ Abstract: "Thanks to previous research that has addressed these issues by proposing a gain-based automated metric called Sem-nCG, which is both rank and semantic aware.": There is no main predicate in this sentence.
+ The citation style looks strange. For example, "ROUGE Lin (2004b)" should be "ROUGE (Lin 2004b)".
+ Equation 2: the font of X is different from the one in the sentence.

**Strengths And Weaknesses:**

# Strengths

The proposed method (a weighted sum of Sem-nCG and redundancy score) is simple and reasonable.

# Weaknesses

**The claims and findings described in this paper need to be revised.**

1) For Table 1, the authors wrote, "the redundancy-aware Sem-nCG metric outperforms the original Sem-nCG metric in terms of Consistency, Relevance, and Coherence; with a 5% improvement in Relevance and a 14% improvement in Coherence for less overlapping references (LOR). We also observe improvements in the Relevance (9%) and Coherence (20%) dimensions for medium overlapping references (MOR). For High Overlapping References (HOR), the improvement is 8% and 22% for Relevance and Coherence, respectively." The problems are:

+ The authors report improvements by comparing a Sem-nCG (without redundancy) score and the maximum value computed from the other four configurations (Cosine, ROUGE, MoverScore, BERTScore). This treatment is problematic because the proposed method has four chances, whereas Sem-nCG has only one.
+ The reported improvements seem to originate from different configurations. This is like cherry-picking, and it's not justifiable for a fair comparison with the baseline method.
    + 5% improvement in Relevance (LOR): Emsemble_sim + BERTScore
    + 14% improvement in Coherence (LOR): USE + BERTScore
    + 9% improvement in Relevance (MOR): STSb-bert + ROUGE or BERTScore (why not Emsemble_sim + BERTScore?)
    + 20% improvement in Coherence (MOR): Ensemble_sim + ROUGE
    + 8% improvement in Relevance (HOR): STSb-bert + BERTScore
    + 22% improvement in Coherence (HOR): Infersent + ROUGE
+ The authors should choose one configuration and compare the values between the proposed and baseline methods.

2) "It also appears that ROUGE and BERTScore provide comparable performances while computing Scorered."

+ I expect to see more concrete evidence of why ROUGE and BERTScore are comparable rather than the conjecture.

3) "In Table 2 Kendall’s tau correlation of ROUGE and BERTScore has been demonstrated to get an idea of the advantage of redundancy-aware Sem-nCG and it is clearly evident that redundancy-aware Sem-nCG also
exhibits stronger correlation than these metrics."

+ This paper needs to explain which BERT model was used for this experiment. This is unfair comparison because the proposed method uses more candidates for embeddings.
+ This claim is incorrect because Relevance (0.20; HOR) and Fluency (0.08; LOR) scores of BERTScore in Table 2 are the best of all configurations in Table 1.

4) "It is evident that λ = 0.5 gives comparable performance in all four quality dimensions."

+ This is not evident at all. Fluency scores continue to increase up to λ = 1.0. A reasonable and detailed explanation of why λ = 0.5 is the best is necessary.

**This paper does not fully explain the proposal of how multiple reference summaries are used in the proposed method.** I can see that P8 includes a description, but it should be presented in Section 3.

**The presented method is simple but too incremental.**

---

> ### Author Response · Authors · 2023-10-05
>
> * The authors report improvements by comparing ... the proposed and baseline methods.
>   * We would like to clarify our approach and address the concerns raised. We experimented with multiple configurations to explore the effectiveness of different metrics for redundancy scores in combination with the Sem-nCG metric. To ensure a fair comparison, we maintained the same settings as the original Sem-nCG while creating the groundtruth ranking. Different approaches have been explored to compute score_red. Based on our analysis, there are no general winners and we have made some recommendations based on the analysis. Quoted from the paper “We also observe that STSb-distilbert embedding is a better choice in the Consistency dimension, whereas USE with enc-2 is a better choice in the Relevance and Coherence dimensions to construct the groundtruth ranking. Therefore, we recommend STSb-distilbert to create groundtruth ranking if Consistency is a top priority, otherwise, we recommend using USE with enc-2. A groundtruth ranking was also created by combining STSb-distilbert and USE into an ensemble, which showed balanced performance across all four dimensions. It also appears that ROUGE and BERTScore provide comparable performances while computing Scorered. However, using ROUGE score as self-referenced redundancy will be a better choice as evident from Section 5.3.” We plan to include a summary table in the revised version to clarify the observations.
>
> * I expect to see more concrete evidence of why ROUGE and BERTScore are comparable rather than the conjecture.
>   * You are right. we have not provided any concrete evidence. We made these observations based on the analysis, but we're not sure what other experiments we can run here. Could you kindly recommend any experiments?
>
> * This paper needs to explain which BERT model was used for this experiment. This is unfair comparison because the proposed method uses more candidates for embeddings.
>    * This is not unfair to our best understanding, we deliberately did that to see if there are any embeddings available to capture the redundancy of model’s summary. We kept the setting of the original Sem-nCG metric as it is; instead, we enhanced it by incorporating a redundancy term in the model's summary, as described in our methodology. We're also not sure which experiment you're referring to when you ask about the BERT model that was used; to create groundtruth ranking, we used three different bert-based embeddings, including STSb-bert, STSb-roberta, and STSb-distilbert (we mentioned that in Section 4.2.1). On the other hand, if you meant about BERTScore for the redundancy penalty, we used "bert-base-uncased" with the default BERTScore settings.
>
> * "It is evident that λ = 0.5 gives comparable performance in all four quality dimensions." This is not evident at all. Fluency scores continue to increase up to λ = 1.0. A reasonable and detailed explanation of why λ = 0.5 is the best is necessary.
>   * We would be happy to rephrase our findings, fluency makes more sense for abstractive summarization since it primarily pertains to grammatical correctness according to the definition. If we ignore fluency for extractive summarization, as extractive sentences are inherently grammatically correct. For a single metric, balancing the diverse quality dimensions—coherence, consistency, fluency, and relevance—requires careful consideration. In our analysis, we observed that a balanced λ value of 0.5 yielded comparable performance across all four quality dimensions. This suggests that this configuration strikes a reasonable tradeoff between relevance and diversity, addressing the complexities inherent in assessing summarization quality comprehensively from a single score provided by the metric. From our observations as mentioned in the paper “For both redundancy penalties, it shows that higher lambda (λ ≥ 0.6) achieves better correlation for the Consistency dimensions, which makes sense because higher lambda means giving more weight to Sem-nCG. For Relevance and Coherence dimensions, a lower lambda (λ) value between [0.3 − 0.5] is a better choice as lower λ means more penalty to redundancy. It appears that for Fluency all metric variations struggle. It is evident that λ = 0.5 gives comparable performance in all four quality dimensions (consistency, relevance, coherence and fluency) and thus we recommend using λ = 0.5 while adopting Equation 3 to compute redundancy-aware Sem-nCG.”

---

### Review · Reviewer_j4av · 2023-09-29

**Summary Of Contributions:**

This paper proposes a way to improve an existing metric Sem-nCG, such that it can handle two additional requirements that are not covered by the original metric: multi-reference evaluation and redundancy in summary. The proposed solution is a linear combination of the original Sem-nCG and the self-referenced redundancy score proposed by Chen et al. (2021). Experiments and further analysis (e.g., hyper-parameter choices) demonstrate the value of the proposed metric.

**Audience:**

Yes

**Broader Impact Concerns:**

No concern about the ethical implications.

**Claims And Evidence:**

No

**Requested Changes:**

- More exploration on different strategies of combining the two components.
- More evaluation datasets in the experiment section
- Addressing the format issues if possible
- Further explanation on why solving the two issues together (multi-reference evaluation and redundancy) is not trivial
- Further explanation on the technical contribution on multi-reference evaluation, and why the original Sem-nCG cannot do that

**Strengths And Weaknesses:**

**Strengths**

- This work proposed an improved version of Sem-nCG, which can handle both multi-reference evaluation and redundancy in summary.
- The experiment with the CNN/DailyMail dataset is exhaustive and provides sufficient evidence for readers to understand this metric.

**Weaknesses**

There are several weaknesses in this work and addressing them will make it more solid in my opinion.

- The argument of the challenges is not convincing. This work points out that addressing two challenges (multi-reference evaluation and redundancy) together is non-trivial. However, it does not provide further evidence about why it is challenging.
- Furthermore, the technical contribution explained in section 3 does not involve any strategy to address the multi-reference evaluation, which makes me further wonder what exactly the challenge claimed in the introduction section.
- The combination of two components in equation (3) makes sense to me. However, I am still wondering whether this is the best option. Unfortunately, this work does not answer this question directly.
- Figure 1 shows some promising results when combining with redundancy penalty, however there is no consistent pattern. Choosing an evaluation metric, unlike choosing a model, we have to pre-select a configuration that works for every scenario. Unfortunately, I am not sure this is the case here.
- Although it’s very impressive that the experiments are very through on the CNN/DailyMail dataset, to make a convincing case, I think it’s reasonable to ask for the results on some other benchmark datasets, unless there is a fundamental reason of not doing that.

Last, there are some format issues, such as the citation format in text and word capitalization in the middle of sentences.

---

> ### Author Response · Authors · 2023-10-05
>
> We sincerely appreciate your thoughtful comments and helpful feedback.
>
> * The argument of the challenges is not convincing ...
>    * Please give us the opportunity to rephrase our claim. Multi-reference evaluation of summarization is challenging because humans have different variability, focus, and writing styles while writing reference summaries, now how can we create a single groundtruth ranking based on this variability is challenging? On the other hand, in extractive summarization systems can extract different important sentences, while concatenating these important sentences which may make the summary redundant. So incorporating redundancy into the metric is also challenging while keeping a balance between relevance and diversity. What we meant, proposing a metric, that incorporates these two different challenges together is non-trivial.
>
> * Furthermore, the technical contribution ...
>     * As mentioned earlier, multiple reference summaries may have variability, now creating a robust groundtruth ranking of sentences from multiple references for comparison is challenging. We have gone through an intuitive approach here, creating different groundtruth rankings based on different references and then concatenating these groundtruth rankings based on average cosine similarity and relevances to make them robust across multiple references. Right now, we only discuss that in the experimental section (Section 5.3).  We would be happy to address this in the methodology section in the revised version (if accepted).
>
> * The combination of two components ...
>     * Thank you for your thoughtful comment. This is a valid point, there are many different possibilities but we went for an intuitive approach that is explainable as well while combining these two options. Other alternatives are definitely possible and while working we have also thought about this. For example, another option can be creating a groundtruth ranking that will be redundancy-aware, explanations and reasons we avoid this approach are below:
>
> Computational Efficiency: The MMR algorithm is another classic approach that handles redundancy, in its conventional form MMR, selects a specific sentence and then compares all other sentences to the selected one, choosing the least similar sentence. This process continues iteratively. While this approach is effective, it becomes computationally inefficient when dealing with a large number of sentences (>3). In the ground truth ranking creation for Sem-nCG, the documents can have N (N is on average 35 for CNN/DM) sentences. Creating a groundtruth ranking with N sentences would require an extensive number of comparisons (first comparing with N-1 sentences, then comparing the next two selected sentences with the remaining N-2 sentences, and so on).
>
> Ambiguity in Redundancy: The second challenge arises from the ambiguity in determining redundancy. In a document with multiple sentences, there can be different groups of sentences that are ideally suited as top sentences for groundtruth ranking. For example, in a document with sentences s1, s2, s3, ..., s20. Assuming s1 and s2 are paraphrases of each other, Sentence sets like(s1, s3, and s5)  or (s2, s3, and s5) could both be considered good choices where the group with (s1, s2 and s5) is redundant. This variability makes it challenging to establish a single, definitive groundtruth ranking.
>
> Given these considerations, we opted for a more straightforward approach, which involves creating the groundtruth ranking of sentences based on the overall similarity of reference sentences and then applying a redundant penalty to the model's extracted sentences, that is intuitive and explainable.
>
> * Figure 1 shows some promising results ...
>
>    * We genuinely appreciate your thoughtful comment. In the pursuit of a single evaluation metric that can account for different multi-dimensional qualities, we acknowledge the inherent challenge. Balancing these diverse quality dimensions—coherence, consistency, fluency, and relevance—requires careful consideration. If we exclude fluency from the hyperparameter choice since it primarily pertains to grammatical correctness, extractive sentences are inherently grammatically correct. In our analysis, we observed that a balanced λ value of 0.5 yielded comparable performance across all four quality dimensions. This suggests that this configuration strikes a reasonable tradeoff between relevance and diversity, addressing the complexities inherent in assessing summarization quality comprehensively with a single score from the metric.

---

> ### Author Response · Authors · 2023-10-05
>
> * Although it’s very impressive that the experiments are very through ...
>    * We used the dataset by Fabbri et al. (2021), the only available benchmark "meta-evaluation dataset" for extractive summarization, to the best of our knowledge. Sem-nCG's authors have demonstrated its correlation with human judgment on this dataset. To ensure a fair comparison, we maintained the same settings as the original Sem-nCG when assessing the redundancy-aware Sem-nCG. To evaluate the redundancy-aware Sem-nCG we will need a similar kind of evaluation benchmark and we can not do anything here. Even though “Liu et al. Revisiting the Gold Standard: Grounding Summarization Evaluation with Robust Human Evaluation. ACL (1) 2023” has published a new dataset, that work focuses mainly on how to increase human annotation reliability for summary evaluation with respect to Atomic Content Unit (ACU) and doesn’t provide human judgment for model’s summary along four summary quality dimensions: coherence, consistency, fluency and relevance.
>
> * Further explanation on the technical contribution on multi-reference evaluation, and why the original Sem-nCG cannot do that
>
>   * The original Sem-nCG has shown how to create groundtruth ranking from 1 reference only. However, how can we create only 1 groundtruth ranking when multiple reference summaries are available has not been answered in the original Sem-nCG work. When multiple reference summaries are available, we can create different groundtruth rankings from different references and these rankings will vary as different reference summaries for the same sample can vary because of the variability of focus and writing style in reference summaries. We have shown a detailed analysis of how to ensemble these multiple rankings into one based on similarity and relevance scores (Section 5.2).

---

### Decision · Action_Editor_kbzx · 2023-10-30

**Recommendation:** Reject

**Comment:**

As discussed under "claims and evidence", the reviewers all agreed that this submission did not meet the bar for publication. In addition, reviewers all felt that the contribution was relatively small (though this is not a criterion for acceptance/rejection at TMLR).

**Audience:**

I think there are likely at least a few people in TMLR's audience who care about evaluating extracting summarization, though it is a very specialized topic.

**Claims And Evidence:**

The reviewers did not feel that the claims in the paper were supported by evidence. In particular,
- Reviewer iEg2 felt that overall the experiments did not support the hypothesis in the paper, in particular due general preference to the proposed method (e.g. different experimental settings considered across methods)
- Reviewer j4av felt that design choices were not sufficiently justified.
- Reviewer Y35R felt that the experiments should have been run on pre-trained Transformers and that the fact that the method allowed multiple choices of embeddings conflated results.